# Cashew By-Product as a Functional Substrate for the Development of Probiotic Fermented Milk

**DOI:** 10.3390/foods12183383

**Published:** 2023-09-09

**Authors:** Marcos Edgar Herkenhoff, Igor Ucella Dantas de Medeiros, Luiz Henrique Grotto Garutti, Mateus Kawata Salgaço, Katia Sivieri, Susana Marta Isay Saad

**Affiliations:** 1Department of Biochemical and Pharmaceutical Technology, School of Pharmaceutical Sciences, University of São Paulo (USP), Av. Professor Lineu Prestes, 580, São Paulo 05508-000, SP, Brazil; marcos.herkenhoff@gmail.com (M.E.H.); igorucella@gmail.com (I.U.D.d.M.); luizgarutti@gmail.com (L.H.G.G.); 2Food Research Center FoRC, University of São Paulo (USP), Av. Professor Lineu Prestes, 580, São Paulo 05508-000, SP, Brazil; 3School of Pharmaceutical Sciences of Araraquara, São Paulo State University (UNESP), Rodovia Araraquara Jaú, Km 01 s/n, Araraquara 14800-903, SP, Brazil; mateus.salgaco@unesp.b (M.K.S.); katia.sivieri@unesp.br (K.S.)

**Keywords:** probiotics, prebiotic, by-product, antioxidant activity, fermentability assay, in vitro digestion

## Abstract

Cashew (*Anacardium occidentale*) processing generates a by-product (CB) with potential for health benefits and that could be a favorable ingredient to be added to a probiotic food matrix. This study aimed to assess the functional attributes of CB in fermented milk with a probiotic and a starter culture using in vitro gastrointestinal conditions. Two formulations were tested, without CB (Control Formulation—CF) and with CB (Test Formulation—TF), and the two strains most adapted to CB, the probiotic *Lacticaseibacillus paracasei* subsp. *paracasei* F19^®^ and the starter *Streptococcus thermophilus* ST-M6^®^, were chosen to be fermented in the CF and the TF. During a 28-day period of refrigeration (4 °C), both strains used in the CF and TF maintained a population above 8.0 log CFU/mL. Strains cultured in the TF had a significant increase in total phenolic compounds and greater antioxidant potential during their shelf life, along with improved survival of F19^®^ after in vitro-simulated gastrointestinal conditions. Our study revealed the promising potential of CB in the probiotic beverage. The CB-containing formulation (TF) also exhibited higher phenolic content and antioxidant activity. Furthermore, it acted as a protector for bacteria during gastrointestinal simulation, highlighting its potential as a healthy and sustainable product.

## 1. Introduction

Cashew (*Anacardium occidentale*) is a non-climacteric fruit that is traditionally endemic to northeastern Brazil. The fruit is composed of both a true fruit, which is the chestnut, and a pseudofruit, which is the pulpy part of the apple. It is commonly used by the Brazilian culinary population in the production of sweets, juices, and, primarily, dried and/or roasted chestnuts. However, due to the high economic value of the chestnut, a significant amount of pseudofruit is wasted during processing. This waste occurs either because of the fruit’s high perishability or after juice extraction, which generates a fibrous matrix with no specific use [1,2]. Preliminary studies have been conducted to explore the potential uses of the pseudofruit in human food [1].

Therefore, the by-products of tropical fruits such as cashews can be regarded as potential sources of dietary fiber and natural pigments, supporting international recommendations to increase fiber consumption [3,4,5]. Compounds found in the peel, seeds, and residual pulp of fruits can provide energy for the intestinal microbiota, leading to the generation of several bioactive metabolites with increased bioavailability and/or bioactivity [6]. Fruits and their by-products possess potential for application in a variety of food and pharmaceutical products as well as in the environmental sector, owing to their diverse chemistry [7]. They encompass a substantial quantity of macronutrients (carbohydrates, proteins, and fats) and micronutrients (vitamins and minerals). Furthermore, they are rich in bioactive compounds such as carotenoids, sterols, phenolic compounds (including flavonoids and non-flavonoids), and dietary fiber [8]. Several studies have documented the potential benefits of fruit by-products, including anti-aging effects [9], antioxidative properties [10], anticancer attributes [11], antidiabetic potential [12], and neuroprotective qualities [11]. It is important to highlight that the gut microbiota plays a crucial role in nutrient synthesis, digestion and absorption, immune modulation, pathogen control, modulation of energy extraction from food, and regulation of appetite [13].

In addition to improving the gut microbiota, the use of fruit by-products as potential prebiotic ingredients that are selectively utilized by beneficial allochthonous microorganisms has been studied [14,15]. Prebiotics are substrates that can “stimulate the growth of beneficial microorganisms, which confer health benefits to the host” [16]. In contrast, probiotics are “live microorganisms that confer a health benefit on the host when administered in adequate amounts” [17], with many of them belonging to the lactic acid bacteria group [18]. For probiotics, their survival in the food matrix is essential. 

Although there is no consensus on the physiological dose for a probiotic benefit, institutions and researchers have suggested that populations ranging from 6.0–9.0 log Colony-Forming Units (CFU) per gram or serving portion could assist in maintaining healthier gut microbiota [17]. 

Furthermore, fruit by-products may contain various phenolic compounds, including those of the flavonoid type (catechins, quercetin, procyanidin dimers, and proanthocyanidins) and those of the non-flavonoid type (phenolic acids, stilbenes, tannins, coumarins, etc.) [19]. The diverse chemical composition of fruit by-products, including cashew pseudofruit, could assure their significant prebiotic potential and be a food matrix for probiotics. Although probiotics share several common characteristics, as every fruit and its by-product represent intricate matrices, their fermentation dynamics can exhibit significant variations, even within the same matrix. Thus, fermentability assays are indispensable for selecting optimal probiotics and starter cultures. Fermented milks, which contain lactic acid, are products resulting from the fermentation of numerous bacterial genera, either as primary or secondary fermentation. Among these genera, the term lactic acid bacteria is specifically designated for the genera within the order Lactobacillales, encompassing *Lactobacillus* (including former members of this genus that have recently been reclassified) and *Streptococcus* [20].

Among the bacteria within these genera, the probiotic strain *Lacticaseibacillus paracasei* subsp. *paracasei* F19^®^ (F19) and the starter culture *Streptococcus thermophilus* ST-M6^®^ (ST-M6) stand out. F19 is a strain that maintains stability throughout the production process [21,22,23], offers established health benefits [24], and demonstrates compatibility when co-fermented with *Streptococcus thermophilus* starter strains [23]. In a study involving passion-fruit by-products, ST-M6 exhibited enhanced folate production and the ability to increase folate levels in soymilk formulations, both independently and in conjunction with *Lactobacillus* spp. strains [25].

Therefore, this study aimed to assess the functional aspects of an isolated cashew by-product (CB) and its application in fermented milk with a probiotic strain and a starter culture and to verify the survival of the inoculated strains in a static in vitro model. To achieve this objective, the dehydrated CB was characterized in terms of its phenolic compounds and antioxidant activity. A probiotic strain and a starter strain were selected for application in the fermented milk, based on fermentation in the presence and absence of CB. The survival of probiotic bacteria under in vitro gastrointestinal conditions was evaluated in fermented milk with and without CB.

## 2. Materials and Methods

### 2.1. Cashew By-Product (CB)

The experimental schematic diagram is outlined in Figure 1. Approximately 15 kg of cashew by-product (CB) was obtained from a local juice processing industry in Natal, Rio Grande do Norte, Brazil. The collection, storage, and transportation to São Paulo, Brazil, where it was dehydrated, were performed with this matrix frozen at −18 °C. First, portions of 500 g were hermetically packed and heated in boiling water for 2 min, followed by an ice bath cooling. Next, they were dehydrated to constant weight at 60 °C in an oven with air circulation (Solab, São Paulo, Brazil), milled, and sieved to obtain flour with an approximate particle size of ≤0.42 mm, as described by Praia et al. [21], and stored in hermetically closed glass bottles under light protection at −20 °C. A 10 g portion of the dehydrated CB, kept for the fermentability study, was sterilized by irradiation in Gammacell 220 (Atomic Energy Canada Ltd., Ottawa, ON, Canada), with an activity of 1287.6 Ci and dose of 25 kGy at a rate of 1.089 kGy/h, according to Albuquerque et al. [8]. 

The non-irradiated CB was evaluated for its antioxidant activity and total phenolic compounds using the Folin–Ciocalteu reagent, DPPH, and gallic acid (Sigma-Aldrich, St. Louis, MO, USA). The characterization of the phenolic compound profile of CB was also conducted.

### 2.2. Fermentability Assay of Probiotic and Starter Bacteria in CB

Cashew by-product (CB) was evaluated for its ability to interfere with the growth of a range of bacteria using in vitro fermentability as described by Vieira et al. [15]. A total of 10 probiotic strains and 3 starter strains were selected (Table 1) to achieve an inoculum of 4–5 log CFU/mL, determined by traditional plating methods as described below.

The frozen strains were thawed and subsequently reactivated in 5 mL of specific broth under specific cultivation conditions (Table 1). Aliquots of 100 µL were added to 5 mL of modified MRS broth—MRSm supplemented with 1% of CB previously sterilized by irradiation, as described above—followed by incubation at 37 °C for up to 48 h.

The microorganisms in MRSm + 1% CB were enumerated by colony counting on selective agar at the initial inoculum (0 h) and after 24 h and 48 h of incubation (Table 2). Control tests were also conducted to verify the effectiveness of CB sterilization and the fermentative capacity of all strains in pure MRSm broth (without CB). The results were expressed as the variation in microorganisms’ populations (delta—Δ) between 0 and 24 h of fermentation (Δ24) and between 0 to 48 h of fermentation (Δ48) (Table 2). Based on these population variation results, combined with scientific research on the tested strains, the probiotic strain *Lacticaseibacillus paracasei* subsp. *paracasei* F19^®^ and the starter strain *Streptococcus thermophilus* ST-M6^®^ were selected for use in the fermented milk formulations.

### 2.3. Cultures Employed and Fermented Milk Production

The probiotic strains *Lacticaseibacillus paracasei* subsp. *paracasei* F19^®^ (F19) and the starter culture *Streptococcus thermophilus* ST-M6^®^ (ST-M6), both from Chr. Hansen (Hørsholm, Denmark), were chosen for use due to their known abilities and survival rates in various food matrices. The probiotic and the starter strains, frozen probiotics, and starter cultures (preserved at −80 °C with 20% glycerol) were activated at 37 °C for 24 h in the specific broth for each strain. F19 strains were activated in MRS (De Man, Rogosa and Sharpe, Oxoid) broth under anaerobic conditions with the AnaeroGen^TM^ Anaerobic System (Oxoid), and ST-M6 strains were activated in Hogg–Jago Glucose (HJ) broth under aerobic conditions, following the methodology described by Battistini et al. [22].

For the formulation of fermented milk, partially skimmed (2.4% milk fat) lactose-free sterilized powdered milk (Ninho^®^, Nestlé, Vevey, Vaud, Switzerland) was used (Table 3). The reconstituted milk was heated with agitation in a food processor (Thermomix, Vorweck^®^, Germany) with 5% sugar (União^®^, Brazil), 2.5% powdered milk (Ninho^®^, Nestlé, lactose-free), and 0.1% unflavored gelatin (Royal^®^, Mondelez). A total of 2.5% of non-sterilized cashew by-product (CB) was added only to the Test Formulation (TF), and, along with the Control Formulation (CF), they were both pasteurized for 5 min at 90 °C and cooled in an ice bath to 37 °C. Finally, the probiotic and starter cultures were added, resulting in a pre-fermentation inoculum of at least 8.0 log CFU/mL of each strain.

The formulations were incubated at 37 °C until they reached pH 5.5 and refrigerated at 4 °C overnight. After overnight, the fermented milk formulations were homogenized with the aid of sanitized spatulas. Portions of 25 g were subdivided into polypropylene cups (Tries Aditivos Plásticos, São Paulo, Brazil), sealed, and stored under refrigeration (4 °C) for up to 28 days. 

### 2.4. Evaluation of Microorganism Survival during Storage

Three batches of each formulation were produced, and the evaluation of their shelf life was conducted from the first day after obtaining the product and weekly on days 7, 14, 21, and 28 of storage, using the parameters described by Battistini et al. [22] as follows. The viability of probiotic and starter microorganisms in the 3 batches of fermented milk was determined during refrigerated storage in the intervals cited above for both formulations. Three portions of 10 g (taken from independent plastic pots) were mixed with 90 mL of sterile saline solution (0.85%), followed by subsequent serial dilutions. The microbial counts were expressed in log CFU/g using the pour plate technique, in which 1 mL of each dilution was thoroughly mixed with the appropriate agar culture medium for each species (Table 1).

### 2.5. Physicochemical Characterization

The chemical composition of the cashew by-product (CB) and the fermented milk formulations (CF, Control Formulation; TF, Test Formulation) was determined, following recommendations from the Official Methods of Analysis of the Association of Official Agricultural Chemists (2012). The following parameters were determined: moisture (925.09—AOAC 2012), fixed mineral residue (930.30—AOAC 2012), total fats [26], total proteins (990.03—AOAC 2012), and total and insoluble dietary fibers (respectively, 985.29 and 991.42—AOAC 2012). 

The available carbohydrate content was calculated by the difference. The fiber content of the TF was estimated based on the amount of CB added in the Test Formulation (2.5%). Direct pH measurement was performed in triplicate on the CB solution, CF, and TF with Orion Three Stars equipment (Thermo Fisher Scientific, Waltham, MA, USA) using a penetration electrode (model 2A04-GF (Analyzer)). The CB acidity was quantified using the titration method for expression in equivalents of malic acid (IAL, 2008). For this, the measurement was carried out in a solution of 10 g in 100 mL of distilled water homogenized in BagMixer 400 (Inter Science, St. Nom, France) for 2 min (IAL, 2008).

The CB was subjected to 2 consecutive extractions under stirring at room temperature in a ratio of 1:20 with 70% methanol. The extract was subjected to analysis to detect total phenolic compounds using the Folin–Ciocalteu reagent (Sigma-Aldrich, Barueri, Brazil) and DPPH to determine the free radical antioxidant potential [5]. The methanolic extraction of CB was subjected to a solid-phase extraction with CC 6 polyamide columns (1 g/6 mL) (Macherey-Nagel Gmbh and Co., Duren, Germany) conditioned with methanol and distilled water. The methanol and methanol–ammonia were extracted, and the CB was completely evaporated under reduced pressure at 40 °C, dissolved in methanol, and filtered through a 0.22 µm filter membrane (polytetrafluoroethylene—PTFE; Millipore).

Phenolic compounds of CB were profiled by reversed-phase HPLC (Hewlett-Packard 1100 system) with an autosampler and a quaternary pump coupled to a diode array detector. The eluent solvents were A (water/tetrahydrofuran/trifluoroacetic acid 98:2:01) and B (acetonitrile). Extracts were monitored at 270 nm and injected in triplicate for each eluate (methanol and methanol/ammonia). To identify the phenolic compounds of interest, the peak identification, retention times, and spectral characteristics of the diode matrix were compared to standards and spectra from the database.

### 2.6. Evaluation of Fatty Acid Profile

Lipids extracted from cashew by-product (CB), Test Formulation (TF), Control Formulation (CF) samples were submitted to a cold extraction of total lipids [27] and detection of the fatty acid profile, followed by the esterification of fatty acids with methanolic boron trifluoride and quantification by gas chromatography coupled to mass spectrometry (GC-MS, Agilent 7890a GC, Santa Clara, CA, USA) with C23:0 methyl trichosonoate as an internal standard. Values were expressed as a percentage.

### 2.7. Viability of Microorganisms under In Vitro-Simulated Gastrointestinal Conditions

This parameter was investigated in fermented milk samples at four storage times (days 1, 7, 14, and 28). For this purpose, in vitro-simulated gastrointestinal conditions adapted by Battistini et al. [22] were used to assess the survival of the probiotic and starter strains. The samples were diluted 1:10 in sterilized peptone water (0.1%), and aliquots of 10 mL were passed through three sequential phases: Gastric Phase (Gast) and Enteric Phases 1 and 2 (Ent1 and Ent2). The passing of samples through each of the phases was performed for 2 h in a metabolic bath (MA-095, Marconi, Brazil) at 37 °C and 150 rpm of agitation with the following parameters: Gast at pH 2.0–2.5 with HCl and pepsin and lipase at concentrations of 3 g/L and 0.9 mg/L, respectively; Ent1 at pH 4.5–5.5 with buffer solution and bile salts and pancreatin at concentrations of 10 g/L and 1 g/L, respectively; and Ent2 at pH to 6.5–7.5 and the same concentrations of bile salts and pancreatin as in the previous phase. 

The counts of probiotic and starter cultures were obtained with aliquots collected from triplicates at time 0 and after each phase. Aliquots of 1 mL of serial dilutions in sterile buffered peptone solution (0.1%) were used in the phases that simulated gastrointestinal conditions (and sterile 0.85% saline solution for time 0). Aliquots were submitted to counting using the pour plate technique in specific agar and incubation conditions (Table 1). Results were expressed in log CFU/g of each fermented milk formulation.

### 2.8. Statistical Analysis

All statistical analyses were performed using the Minitab statistical package, version 17.3.1 for Windows (Minitab Inc. 2013, State College, PA, USA). The results were submitted to descriptive analysis as mean, standard deviation, or standard error of the mean. For non-parametric data, the median was presented, followed by the first and third quartiles.

Levene’s tests were performed to determine the equality of variances and Shapiro–Wilk’s tests to evaluate normal distribution. Depending on these results, parametric or nonparametric tests with a significance level of 95 or 99% were used, followed by the appropriate mean or median comparison. If necessary, data that did not assume a normal distribution were subjected to Johnson’s transformation and, in this case, if there was no equality of variances, the means were paired and compared using the Games–Howell test with a significance level of 95 or 99%.

## 3. Results and Discussion

### 3.1. Physicochemical and Functional Characteristics of the Cashew By-Product (CB)

A high protein content profile was observed in the cashew by-product (CB), with 10.67 (±0.02) g of protein per 100 g of by-product (Table 4), which is higher than certain cereals [28]. Since the evaluated material represents the fibrous portion retained after pressing the pseudofruit, emphasis should be given to the dietary fiber values, with higher values of insoluble fiber expected.

Applying this fibrous matrix as an ingredient can be an important factor in increasing beneficial health effects, such as water retention and satiety (soluble fiber) or the regulation of intestinal effects and increased stool volume (insoluble fraction) [28]. In fact, the total dietary fiber values were higher than those of other by-products, such as apple (55.48 g/100 g), guava (44.30 g/100 g), and acerola (48.60 g/100 g) [5,29]. Regarding pH and acidity, expected values were observed in the fermented milk formulations, due to the origin of CB (pH 3.5–4.5) and titratable acidity in terms of malic acid equivalents (0.22–0.55).

Phenolic compounds in cashew apples tend to be retained in husks, seeds, or residual fibrous pulp [30,31]. Phenolic compounds and dietary fiber from fruits, whether by-products or not, have a positive relationship with the modulation ability of the intestinal microbiota, integrity of the intestinal mucosa, serum levels, and insulin resistance [32,33]. In this study, cashew by-products presented 486.6 mg EAG/100 g of the total phenolic compounds, which directly impacted the increased antioxidant activity as measured by the DPPH free radical assay (88.8% inhibition and IC_50_ of 1.16 mg/mL). These values are higher than those found in apple, banana, and orange by-products [34]. In the current study, it was evident that the high values of total phenolic compounds led to a significant inhibitory capacity. This observation was particularly pronounced in relation to the IC_50_ value. The diversity of results regarding the phenolic compound profile in by-products further underscores the ongoing need to investigate the bioactive potential of these matrices.

Syringic and ellagic acid values exceeded those found in a study with lyophilized by-products of cashew, acerola, and guava, which ranged from 1.0 to 49.0 mg/100 g for ellagic acid and 5.0 to 50.6 mg/100 g for syringic acid [35]. Lower values of ellagic acid were identified in cashew nut extracts [36], and lower values of salicylic acid, vanillic acid, and flavonoids (quercetin, myricetin, naringenin, etc.) were also observed in lyophilized cashew by-products (CB) [5].

The most abundant fatty acids were oleic, palmitic, stearic, linolenic, and linoleic acids (Table 4). CB showed a higher amount of oleic acid than soy, hemp, and lupine, as well as a higher linolenic acid content than that reported for oats [37]. This higher content of oleic acid is related to some of the characteristics of the cashew nut, which also exhibits a higher level of this fatty acid compared to others [38]. The distinctive profile of mono- or polyunsaturated fatty acids in the cashew by-product can be viewed as an additional positive factor in its ability to contribute essential compounds to the development of new foods.

### 3.2. Probiotic Selection

Due to the large number of probiotic strains, the choice of the one most adapted to the cashew by-product (CB) was decided in two stages, Δ48 and Δ24, as described in Table 2. For the Δ48 of fermentation in MRSm + CB, the *Limosilactobacillus reuteri* RC-14^®^ (RC-14) strain showed the highest value (3.32 log UCF/mL), but this was without statistical significance (*p* < 0.05) when compared to *Lacticaseibacillus paracasei* subsp. *paracasei* 431^®^ (431), *Lactobacillus acidophilus* LA-5^®^ (LA-5), *Lacticaseibacillus paracasei* subsp. paracasei F19^®^ (F19) and *Bifidobacterium animalis* subsp. *lactis* BB-12. These five strains with the best Δ48 values were selected for comparison with their respective MRSm Control groups. At this point, it was observed that only strains *L. reuteri* RC-14 and *L. paracasei* subsp. *paracasei* (F19) exhibited significantly higher values (*p* < 0.05) compared with their respective controls. Therefore, further comparisons were conducted exclusively between these strains in terms of Δ24 multiplication. The Δ24 values of these two strains were statistically similar (*p* > 0.05) in MRSm + CB compared to each other and higher (*p* < 0.05) when compared with their respective MRSm Control. However, the objective of the work was to choose only one probiotic strain. 

The *L. paracasei* subsp. *paracasei* (F19) strain has the interesting characteristics of genetic stability throughout production [21,22,23], survival in digestive conditions, beneficial interaction with the intestinal epithelium, and changes in host nutrient utilization and energy metabolism [39]. Other evidence from clinical studies summarized by Jones [40] corroborates more positive claims of *L. paracasei* F19 associated with a high-fiber diet in improving swelling and abdominal pain, as well as the influence of this probiotic in reducing the accumulation of fat by modulating transcription factors in energy metabolism.

In another study with probiotic strains, both RC-14 and F19 showed good adaptability in fruit by-products, okara, or amaranth flour after 24 and 48 h of fermentation [15]. Thus, summing up all this evidence, it was decided to choose the F19 strain for application in fermented milk formulations.

Starter strains were also chosen in stages (Table 2). However, unlike probiotics, the number of starters was considerably smaller. Promising behavior by *S. thermophilus* ST-M6 was observed. This strain had higher Δ48 and Δ24 multiplication values in the MRSm + CB compared with the other strains (*p* < 0.05). When compared with their respective MRSm Control groups, *S. thermophilus* ST-M6 also showed significantly higher values (*p* < 0.05) both in Δ48 and Δ24. Therefore, this strain was chosen to be the starter in the fermented milk formulation.

### 3.3. Viability of Microorganisms Used in Fermented Milk during Storage, pH, and Titratable Acidity

At first glance, it may appear that the presence of CB had little effect on the viability of the microorganism (Figure 2). In both formulations, values above 9.0 log CFU/mL were observed for F19 and above 8.6 log CFU/mL for ST-M6 during 28 days of refrigeration. However, it is normal for the viability of these microorganisms, especially *L. paracasei* and *S. thermophilus*, to exhibit a progressive decrease within up to 27 days of refrigeration in dairy products [41]. Since there was no decrease in our study, this may indicate that the CB could have a protective effect on the strains used. During storage, higher values (*p* < 0.01) were found in the Test Formulation (TF) for the probiotic strain (day 14) and starter strain (day 7), although both increases were lower than 0.5 log CFU/mL. Vieira et al. [42], using probiotic *Lactobacillus* genus in a plant-based beverage with acerola fibers, obtained lower values than those in the present study over a 28-day refrigeration period (minimum of 7.35 ± 0.56 log CFU/mL). 

The ST-M6^®^ strain had a count of 8.8 CFU/mL after 28 days in both the Control Formulation (CF) and the Test Formulation (TF). *S. thermophilus* ST-M6^®^ (ST-M6) showed an increasing trend until day 21, followed by a decline on day 28 in both formulations. The viability of probiotic microorganisms following refrigerated storage in both the CF and the TF was acceptable. International institutions such as those in Canada [43] consider amounts of more than 9 log CFU/portion of the more traditional probiotics (*Lactobacillus* spp. or *Bifidobacterium* spp.) adequate to claim a general benefit in improving the health of the intestinal microbiota [17]. The viability of F19 remained higher than that of ST-M6 throughout the storage period, reaching 9.3 log CFU/mL on day 28 for the CF and the TF. This could be attributed to the higher initial inoculum of the probiotic strain compared with the starter strain. Despite this observation, the starter strain exhibited a tendency to increase up to day 21, followed by a decline on day 28 in both formulations. Statistically significant differences (*p* < 0.05) were found over the shelf life studied, yet the microbiological relevance can be considered in limited scenarios. Employing a significance level of 95%, the inherent methodological imprecisions are likely to prevail. In this context, the microbiological significance would be more considerable when differences equal to or exceeding 0.5 log CFU/mL are quantified.

The active metabolism of microorganisms present in the fermented milk also resulted in acidification during their storage (Figure 2) (*p* < 0.05). At the end of 28 days, free acidity (grams of lactic acid/100 mL) and pH were, respectively, 1.4 and 3.7 for the TF and 1.5 and 3.8 for the CF. It should be noted that both the pH and the titratable acidity of the two formulations were within national and international quality standards for fermented milks throughout their shelf life [44]. 

Vieira et al. [45] observed the same acidification trend during refrigeration in probiotic plant-based beverages with acerola fiber (reaching pH 4.2 in some formulations on day 28). Karaka et al. [46] developed a probiotic yogurt supplemented with 1 or 2% apricot fiber with probiotic strains from the *Lactobacillus* genus and observed similar titratable acidity results to our study after 20 days of storage (1.153–1.265), even though the viability of the strains was at most 7.42 ± 0.14 log CFU/mL at the same time. It is presumed that the dairy matrix and higher initial inoculum of probiotics and starter strains in our study were factors that directly influenced the higher production of lactic acid during refrigeration, consequently leading to the highlighted values of titratable acidity and pH.

### 3.4. Physical–Chemical Composition of Fermented Milk

As expected, there was an increase in total fiber content and, mainly, insoluble fiber content in the Test Formulation (TF) compared with the Control Formulation (CF) (*p* < 0.05) (Table 4). It can be expected that the microbial enzymatic metabolism acts on this added matrix, favoring the production of new derived compounds with an impact on aroma and functionality during fermentation and storage [21,23].

Another point observed was the fatty acid composition (Table 4). CB had little effect on the diversity of these observed compounds, which was like what was observed for dairy bases [47,48]. The CF and the TF showed a predominance of palmitic, myristic, and stearic saturated fatty acids. Yogurt formulations showed equivalence in the values of fatty acids. The intriguing bioactive profile of these fatty acids has demonstrated antiviral and antibacterial activity [41].

Evidence has shown that species from genera such as *Lactobacillus* spp., *Bifidobacterium* spp., and *Streptococcus thermophilus* can interfere with the production of conjugated linolenic acids [49], like rumenic acid, which may have interfered with the TF. Together, rumenic and linolenic acid are compounds with a proven positive influence in reducing the risk of cardiovascular diseases [50]. The profile of mono- and polyunsaturated fatty acids in the current study should also be emphasized, as the presence of CB had a slight impact on the increase in oleic acid and linolenic acid (both identified in CB). 

In cashew by-product (CB), the amount of total phenolic compounds (TPC) was 486.57 ± 10.06 (mg AGE/100 g), which were composed of syringic acid 146.8 ± 7.1, quercetin glycoside 16.8 ± 1.2, catechin 40.4 ± 5.4 and ellagic acid 122.0 ± 1.4. The level of antioxidant activity was 1.16 ± 0.03 (DPPH· IC50) and 88.78 ± 0.13 (DPPH). Total phenolic compounds (TPC) identified by the Folin–Ciocalteu assay were significantly higher (*p* < 0.05) in the TF during refrigerated storage. On the other hand, a significant increase was observed on days 14 and 28 in the CF (Figure 2). The acidification process may have resulted in higher TPC and DPPH values for both formulations due to the influence of the pH in these parameters [51]. 

As for the antioxidant potential measured by DPPH inhibition, a direct proportionality was observed with higher TPC values. Despite a slight oscillation in the percentage of inhibition for the TF on day 14, all values observed during storage were greater than 66% and significantly higher (*p* < 0.05) than those for the CF.

In the present study, the addition of cashew by-product (CB) resulted in better values of antioxidant activity, with a tendency to improve during shelf life. This possible oxidative stability is most likely due to the phenolic compounds [52] naturally present in cashew fiber [53] that are associated with the active metabolism of the microorganisms used. According to Quirós-Sauceda et al. [54], the term “antioxidant fiber” has been employed to characterize “a concentrated dietary fiber containing significant amounts of natural antioxidants associated with non-digestible compounds”.

The presence of microorganisms in the fermented milk should also be highlighted. These microorganisms remained viable and metabolically active throughout the studied shelf life of the fermented milk. There is evidence that the complex substrate of fruit by-products is a target for various enzymes produced by both autochthonous and allochthonous bacteria (including probiotics). Consequently, there is formation of aglycon flavonoids, depolymerization of complex phenolics, and production of short-chain fatty acids or even conjugated polyunsaturated fatty acids [4].

It is hypothesized that the fermentation and consequent acidification of the dairy base, as well as the addition of fibrous matrix from the CB, may have acted synergistically, resulting in the formation and release of metabolites with greater antioxidant capacity. Consequently, the inclusion of F19 may have had a favorable impact in enhancing the antioxidant profile throughout the shelf life of the TF, which leads to a favorable aspect in meeting the requirements for safe foods that utilize natural food additives.

### 3.5. Survival of Microorganisms in Fermented Milk under In Vitro-Simulated Gastrointestinal Conditions

Probiotic and starter strains were evaluated under in vitro-simulated gastrointestinal conditions (Table 5). In the F19 population, viability was always above 9.00 log CFU/mL (F0), although variable survival responses were observed in the formulations stored up to 7, 14, and 28 days under refrigeration. It is interesting to note that the three highest values (*p* < 0.05) of survival under gastric conditions (Gast) were all for the TF on days 7, 14, and 28. Resistance of F19 has already been observed in a static digestion simulation model [22]. 

Considering these periods, the TF was superior to the CF by an average of 2.91 log CFU/mL. Therefore, cashew fiber may be acting as a stress shield against the tension caused by stomach acidity. Moreover, higher populations of surviving F19 microorganisms (*p* < 0.05) were also detected on days 7 and 28 after enteric phase II (Ent2). Thus, dietary fiber may have conferred a potential for bacterial survival under gastrointestinal conditions and improved adhesion to the intestinal epithelium [55,56].

In addition to *L. paracasei* subsp. *paracasei* F19, the survival of the *S. thermophilus* ST-M6^®^ strain was also investigated (Table 5), where a trend toward a progressive decrease in the microorganism population was observed after the gastric phase and enteric phase I and a positive interference by the presence of CB was seen at the end of enteric phase II (*p* < 0.05). It has been demonstrated that the resistance of *Lactobacillus* spp. (including the species formerly known to belong to this genus and now reclassified) to gastric juices might be improved in the presence of citrus fruit and saccharin beet pectins [56]. Furthermore, this viability may also be enhanced in the presence of dietary fibers, reaching levels like or even higher than those achieved with pectins [57]. In addition, the presence of fiber led the bacteria to produce SCFAs, which are favorable for colonic health [57]. 

Dietary fiber encompasses a molecular dynamic that is composed of polysaccharides with the coexistence of soluble and insoluble chains in an aqueous environment. This is evidenced in the CB, where the predominance of insoluble fibers is notable. These fibers have been previously examined under microscopic scrutiny in a study conducted by Medeiros et al. [5], revealing an irregular and porous surface. Consumption of this type of fiber promotes bacterial propagation, wherein bacterial adherence to the substrate plays a pivotal role in their survival and the subsequent effects on the composition and metabolism of the intestinal microbiota [58].

Vieira et al. [45] developed a fermented soy beverage with the co-culture of *Bifidobacterium longum* BB-46^®^, *Lactobacillus acidophilus* LA-5^®^, and *Streptococcus thermophilus* TH-4^®^ enriched with acerola by-product. The authors evaluated the survival of strains under simulated gastrointestinal conditions in the same static model used in this study. Better survival of BB-46^®^ was observed after the gastric and enteric I and II phases (above 5.5 log CFU/mL) during the entire storage period. For the LA-5^®^ strain, on the other hand, survival after the gastric phase and enteric phases I and II was only higher until day 14 of storage. At 28 days, survival was below 3.0 log CFU/mL after all phases. This contrasted with the findings in this study, where the survival of F19 and ST-M6 exceeded 4.4 log CFU/mL, and the presence of CB showed statistical significance (*p* < 0.05) in promoting greater probiotic strain survival compared with the formulation without CB. The presence of dietary fibers can potentially enhance the survival of commensal or beneficial bacteria under gastrointestinal conditions and improve adhesion to the intestinal epithelium [55,56]. Consequently, the likelihood of a sufficient number of microorganisms reaching and remaining viable in the large intestine is increased, enabling them to exert positive effects on intestinal microbiota homeostasis.

## 4. Conclusions

The dehydrated cashew by-product (CB) was revealed to confer to fermented milk favorable features as a food matrix for *Lacticaseibacillus paracasei* subsp. *paracasei* F19^®^ (F19) and *Streptococcus thermophilus* ST-M6^®^ (ST-M6). CB significantly increased the composition of phenolic compounds and antioxidant activity during the storage of the fermented milk formulation. Furthermore, CB proved to be an effective protector for both the starter and the probiotic strains in the static in vitro gastrointestinal simulation assay. The use of CB presents an alternative for the development of healthier beverages, particularly of probiotic fermented milk. Furthermore, it represents a more sustainable option for the environment, as it involves repurposing a waste product. Although survival analyses of probiotics have been conducted in static models simulating gastrointestinal digestion, further studies are required to validate and confirm the probiotic properties of the formulations studied here. Additionally, more studies should be conducted in order to validate their functional aspects in laboratory, animal, and/or human models using this fermented milk formulation with CB.

## Figures and Tables

**Figure 1 foods-12-03383-f001:**
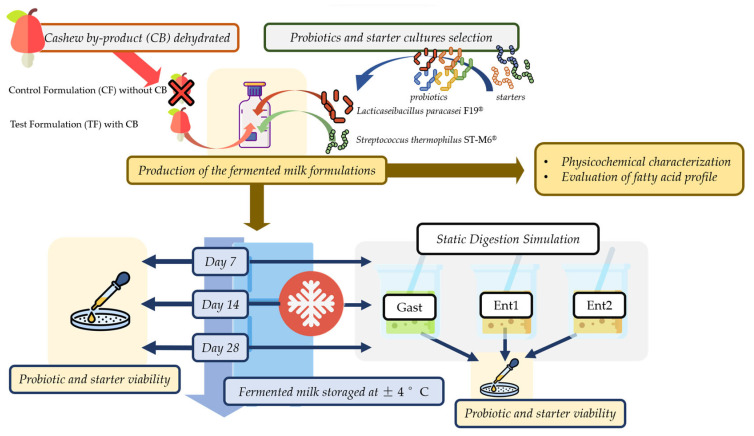
Experimental flowchart of fermented milk formulations with (TF) and without (CF) dehydrated cashew by-product (CB) with the probiotic strain *Lacticaseibacillus paracasei* subsp. *paracasei* F19^®^ and the starter strain *Streptococcus thermophilus* ST-M6^®^.

**Figure 2 foods-12-03383-f002:**
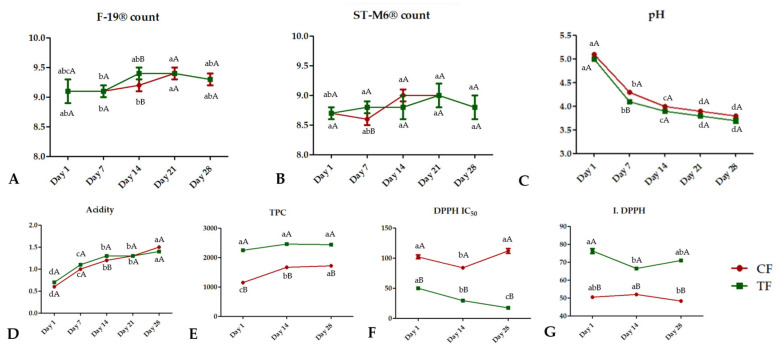
Population (log CFU/mL) variation of probiotic (**A**) and starter (**B**) strains, pH values (**C**), titratable acidity (**D**), total phenolic compounds (TPC) (**E**), and antioxidant activity (**F**,**G**) in fermented milk with (Test Formulation—TF) and without (Control Formulation—CF) the addition of 2.5% of cashew by-product during 28 days of refrigeration. TPC: Total phenolic compounds represented as gallic acid equivalent per 100 g of the formulation (mg EAG/ 100 g). IC50: extraction concentration (mg/mL) obtained from the formulations that can inhibit 50% of DPPH in the reaction. I. DPPH: percentage of DPPH inhibition in reaction (%). Values represented as mean and standard error of the mean. Different lowercase letters in the same line represent significantly different values (*p* < 0.05). Different capital letters for 2 different lines in the same storage period of each parameter represent significantly different values between the CF and the TF (*p* < 0.05).

**Table 1 foods-12-03383-t001:** Microorganisms evaluated in the present study and their cultivation conditions for each strain.

Strain	Broth	Agar	Incubation (at 37 °C)
*Streptococcus thermophilus* TH-4^® a^	HJ ^1^	M17 ^2^	Aerobiosis
*Streptococcus. thermophilus* ST-M6^® a^			
*Streptococcus thermophilus* TA-40^® b^			
*Lactobacillus acidophilus* LA-5^® c^	MRS ^3^	MRS maltose ^4^	Aerobiosis
*Limosilactobacillus fermentum* PCC^® c^	MRS ^3^	MRS ^5^	Anaerobiosis ^7^
*Limosilactobacillus reuteri* RC-14^® c^	MRS ^3^	Acidified MRS ^6^	Anaerobiosis ^7^
*Lacticaseibacillus. paracasei* subsp. *paracasei* 431^® c^			
*Lacticaseibacillus. paracasei* subsp. *paracasei* F-19^® c^			
*Lacticaseibacillus. rhamnosus* GR-1^® c^	MRS ^3^	Acidified MRS ^6^	Aerobiosis
*Lacticaseibacillus. rhamnosus* LGG^® c^			
*Bifidobacterium animalis* subsp. *lactis* BB-12^® c^	MRS cysteine ^8^	LP-MRS ^9^	Anaerobiosis ^7^
*Bifidobacterium. longum* BB-46^® c^			
*Bifidobacterium longum* subsp. *infantis* BB-02^® c^			

^a^: Starter culture (Chr. Hansen, Hørsholm, Denmark). ^b^: Starter culture (DuPont, Dangé, France). ^c^: Probiotic culture (Chr. Hansen). ^1^: Hogg–Jago Glucose (HJ). ^2^: M17 agar (Oxoid, Basingstoke, UK). ^3^: De Man, Rogosa and Sharpe broth. ^4^: MRS agar with maltose (IDF, 1995). ^5^: MRS agar (Oxoid). ^6^: MRS agar acidified to pH 5.4. ^7^: Anaerobic system AnaeroGen™ (Oxoid). ^8^: MRS broth (Oxoid) with L-cysteine (0.05 g/L, Sigma-Aldrich, St. Louis, MO, USA). ^9^: LP-MRS agar. The new taxonomic classification for the previous genus *Lactobacillus* was adopted. All media were used following the recommendations of Battistini et al. [22].

**Table 2 foods-12-03383-t002:** Population (log CFU/mL) variation of probiotic and starter microorganisms after 24 and 48 h of fermentation in MRSm broth with (+CB) and without (Control) cashew by-product.

Strains	Δ48		Δ24	
	MRSm + CB	MRSm Control	MRSm + CB	MRSm Control
*L. reuteri* RC-14	3.32 ± 0.13 ^aA^	2.57 ± 0.13 ^B^	3.64 ± 0.07 ^aA^	3.27 ± 0.02 ^B^
*L. paracasei* subsp. *paracasei* 431	2.92 ± 0.17 ^abA^	2.79 ± 0.06 ^A^	-	-
*L. acidophilus* LA-5	2.87 ± 0.09 ^abB^	3.14 ± 0.02 ^A^	-	-
*L. paracasei* subsp. *paracasei* F19	2.80 ± 0.10 ^abA^	2.35 ± 0.03 ^B^	3.36 ± 0.22 ^abA^	2.55 ± 0.08 ^B^
*B. animalis* subsp. *lactis* BB-12	2.79 ± 0.33 ^abA^	2.62 ±0.06 ^A^	-	-
*L. rhamnosus* GR-1	2.56 ± 0.08 ^bc^	-	-	-
*L. rhamnosus* LGG	2.56 ± 0.13 ^bc^	-	-	-
*B. longum* subsp. *infantis* BB-02	2.28 ± 0.19 ^bc^	-	-	-
*L. fermentum* PCC	2.23 ± 0.10 ^bc^	-	-	-
*B. longum* BB-46	2.02 ± 0.16 ^c^	-	-	-
*S. thermophilus* ST-M6	3.74 ± 0.12 ^A^	−0.21 ± 0.15 ^B^	0.85 ± 0.13 ^A^	0.11 ± 0.04 ^B^
*S. thermophilus* TA-40	−1.56 ± 0.11 ^A^	−2.19 ± 0.10 ^B^	−0.32 ± 0.04 ^A^	−0.76 ± 0.08 ^B^
*S. thermophilus* TH-04	−0.32 (−0.52–−0.02) ^A^	−2.20 (−2.43–−2.11) ^B^	−0.52 ± 0.03 ^A^	−0.71 ± 0.17 ^A^

Values represent the means and standard deviations or medians and first and third quartiles. Δ48: difference in bacterial multiplication value after 48 h of fermentation expressed in log CFU·mL^−1^; Δ24: difference in bacterial multiplication value after 24 h of fermentation expressed in log CFU·mL^−1^; CB: cashew by-product; MRSm + CB: Modified MRS broth with 1% cashew by-product; MRSm Control: Pure modified MRS broth without cashew by-product. Values followed by different lowercase letters in the same column represent a significant difference according to the Tukey test (*p* < 0.05). Values followed by different capital letters on the same line represent a significant difference between test and control groups (of each Δ24 and Δ48) according to the Student’s *t* test (*p* < 0.05) or Kruskal–Wallis test (*p* < 0.05). Statistical analyses were performed by comparing the strains of each group (probiotics and starters) among themselves.

**Table 3 foods-12-03383-t003:** Ingredients and proportions used in the formulation of fermented milk.

Ingredients	Formulations (g/100 mL of Milk)	
	Test (TF)	Control (CT)
Dehydrated cashew by-product (CB)	2.5	-
Skimmed milk powder without lactose (Ninho^®^, Nestlé, Araras, Brazil)	2.5	2.5
Sucrose (União^®^, Piracicaba, Brazil)	5.0	5.0
Unflavored gelatin (Modelez^®^, Curitiba, Brazil)	0.1	0.1

**Table 4 foods-12-03383-t004:** Physical–chemical composition of the cashew by-product (CB), Test Formulation (TF) with 2.5% cashew by-product and Control Formulation (CF) without cashew by-product.

Parameter or Composite	CB	CF	TF
Humidity g/100 g	2.85 ± 0.11	80.57 ± 0.15 *	78.36 ± 0.27
Ashes g/100 g	1.25 ± 0.02	0.79 ± 0.02	0.83 ± 0.01 *
Proteins g/100 g	10.67 ± 0.02	3.69 ± 0.03	3.74 ± 0.30
Carbohydrates g/100 g	9.52 ± 0.13	10.89 ± 0.15	11.18 ± 0.28
Total fat g/100 g	4.47 ± 0.11	4.06 ± 0.03	4.11 ± 0.23
Palmitic acid (C16:0) µg/g fat	867.6	1449.0 ± 177	1341.0 ± 85.0
Palmitoleic acid (C16:1) µg/g fat	43.8	45.8 ± 8.1	43.8 ± 3.3
Stearic acid (C18:0) µg/ g fat	350.2	691.1 ± 75.3	620.0 ± 38.0
Oleic acid (C18:1 ɷ-9) µg/g fat	1336.6	643.0 ± 109	708.3 ± 61.0
Linoleic acid (C18:2 ɷ-6) µg/g fat	90.2	77.0 ± 13.9	77.0 ± 6.3
Linolenic acid (C18:3 ɷ-3) µg/g fat	54.5	-	8.6 ± 0.9
cis-13-eicosanoic acid µg/g fat	33.4	-	-
Rumenic Acid (C18:2) µg/g fat	-	8.4 ± 1.5	7.8 ± 1.0

Values represented by mean and standard deviation. Carbohydrate value represents the available portion, excluding fibers added. Total, insoluble, and soluble fiber values were estimated in the CF and TF. Parameters marked with * are significantly higher (*p* < 0.05) when compared with each other using the Student’s *t* test (between the CF and TF only).

**Table 5 foods-12-03383-t005:** Population survival (log CFU/mL) of *Lacticaseibacillus paracasei* subsp. *paracasei* F-19^®^ and *Streptococcus thermophilus* ST-M6^®^ during simulated gastrointestinal conditions in the in vitro static system in fermented milk with and without cashew by-product.

Strain	Period	Formulation	F0	Gast	Ent1	Ent2
F19	D7	CF	9.05 ± 0.01 ^aB^	5.28 ± 0.23 ^bD^	2.55 ± 0.06 ^cB^	2.53± 0.08 ^cC^
		TF	9.16 ± 0.08 ^aAB^	8.21 ± 0.17 ^bB^	3.15 ± 0.24 ^dAB^	4.52 ± 0.05 ^cB^
	D14	CF	9.18 ± 0.01 ^aAB^	5.20 ± 0.16 ^bD^	3.67 ± 0.11 ^cA^	3.93 ± 0.16 ^cB^
		TF	9.42 ± 0.09 ^aA^	9.26 ± 0.01 ^aA^	3.66 ± 0.08 ^dA^	4.43 ± 0.11 ^cB^
	D28	CF	9.26 ± 0.01 ^aAB^	4.81 ± 0.09 ^bD^	3.26 ± 0.11 ^dA^	4.19 ± 0.13 ^cB^
		TF	9.23 ± 0.07 ^aAB^	6.55 ± 0.05 ^bC^	3.59 ± 0.06 ^dA^	4.65 ± 0.17 ^cA^
ST-M6	D7	CF	8.70 ± 0.09 ^aA^	5.01 ± 0.08 ^bA^	2.54 ± 0.11 ^dB^	3.55 ± 0.04 ^cB^
		TF	8.81 ± 0.08 ^aA^	5.14 ± 0.07 ^bA^	2.75 ± 0.34 ^cAB^	4.73 ± 0.25 ^bA^
	D14	CF	9.06 ± 0.11 ^aA^	5.07 ± 0.15 ^bA^	3.73 ± 0.08 ^cA^	3.78 ± 0.06 ^cAB^
		TF	8.85 ± 0.22 ^aA^	2.97 ± 0.20 ^cB^	2.52 ± 0.18 ^cB^	4.29 ± 0.10 ^bA^
	D28	CF	8.58 ± 0.31 ^aA^	5.11 ± 0.21 ^bA^	3.58 ± 0.17 ^cA^	4.19 ± 0.16 ^cAB^
		TF	8.52 ± 0.28 ^aA^	5.20 ± 0.16 ^bA^	3.50 ± 0.06 ^dA^	4.49 ± 0.13 ^cA^

Values represented by mean and standard deviation. TF: test formulation with 2.5% dehydrated cashew by-product; CF: control formulation without dehydrated cashew by-product. D7, D14, and D28 represent refrigerated storage days. F0: strain count in fermented milk before simulating gastrointestinal conditions; Gast: gastric phase; Ent1: enteric phase 1; Ent2.: enteric phase 2. Different lowercase letters in the same line represent significantly different values (*p* < 0.05). Different capital letters in the same column represent significantly different values (*p* < 0.05) between the same microorganisms.

## Data Availability

All relevant data and methods are presented in this paper. Additional inquiries should be addressed to the corresponding author.

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
