# Peer review of "Cashew By-Product as a Functional Substrate for the Development of Probiotic Fermented Milk"

_foods, 2023, doi:10.3390/foods12183383_

Round 1

Reviewer 1 Report

The manuscript can be considered for publication after following modifications:

1.       The aim of the study is not explained well, the author needs to present it in a detailed and more clear way.

2.       Many references used in the study are outdated, such as 2003, 2009, 2011 etc. Author needs to cite most recent related references.

3.       The schematic diagram of the experimental setup  should be provided.

4.       Abbreviation should be provided.

5.       Section 2.4. The authors adopted the method from another research, or it is their own. If not, then provide an appropriate reference.

6.       Table 4 results can be presented in a graph for better understanding.

7.       The conclusion section should be re-written with future recommendations.

Author Response

We thank the reviewers for their comments and suggestions. We are submitting a revised version of the manuscript with substantial modifications. In order to facilitate the visualization of changes and to ensure that each reviewer can easily identify modifications related to their suggestions, we have decided to mark the revisions suggested by Reviewer #1 in green and the ones suggested by Reviewer #2 in blue.

Reviewer 1

The manuscript can be considered for publication after following modifications:

  1. The aim of the study is not explained well, the author needs to present it in a detailed and more clear way.

  1. We acknowledge the reviewer's comments. Consequently, we have expanded our explanations regarding the objectives of this study at the end of the introduction.

  1. Many references used in the study are outdated, such as 2003, 2009, 2011 etc. Author needs to cite most recent related references.

  1. After a careful evaluation and verification of the references, we have removed or replaced those that are unnecessary, especially the older ones. Approximately 12 references have been removed or substituted. The ones that have been replaced are indicated in green. However, despite their age, Hill et al. 2014, Bligh and Dyer, 1959, and Gholami et al. 2014, could not be removed due to their significance.

  1. The schematic diagram of the experimental setup should be provided.

  1. For a better comprehension, we have generated a figure illustrating the experimental setup of the study.

  1. Abbreviation should be provided.

  1. We agree that a list of abbreviations would be convenient and greatly facilitate reading for the reader. However, the structure of a Foods article does not accommodate an abbreviations list. Unfortunately, we are unable to fulfill this request. However, we included the complete name following its respective abbreviation at the beginning of each section instead.

  1. Section 2.4. The authors adopted the method from another research, or it is their own. If not, then provide an appropriate reference.

  1. The reference has been added in this section.

  1. Table 4 results can be presented in a graph for better understanding.

  1. As recommended, we have converted Table 4 into Figure 2.

  1. The conclusion section should be re-written with future recommendations.

  1. We have rewritten the conclusion as per the request of both reviewers, and we have highlighted these modifications in yellow.

Reviewer 2 Report

My comments are as follow:

·         The abstract section needs to be improved, specially at the end, with more accurate information about the conclusion of your research. Moreover, the conclusion section of the manuscript is very similar that the abstract, therefore, please modify them.

·         In line 29, in the keywords section. Since some words are repeated in the manuscript title and in the keyword section, I suggest to replace the words: “probiotics”, “by-product”, cashew”, or “fermented milk”, by “antioxidant activity”, “fermentability assay”, “physicochemical characterization”, “fatty acid profile” or other keywords, to have more visibility of your manuscript paper.

·         In the Introduction section, please, improve it including more information related to: i) the diverse chemical composition of fruit by-products; ii) fermentability assay, and iii) cultures employed, and the fermented milk production.

·         In the Statistical Analysis section, did you the Levene's tests were performed for equality of variances and Shapiro-Wilk's tests to evaluate normal distribution. Nevertheless, what about the analysis of “autocorrelation in residuals”? Did you apply it? Why not?

·         Please, improve the discussion section of all your results including more information related to: i) Physical-chemical and functional characteristics of the cashew by-product, ii) Probiotic selection, iii) Viability of microorganisms used in fermented milk during storage, pH and titratable acidity, iv) Physical-chemical composition of fermented milk, and v) Survival of microorganisms in fermented milk under in vitro-simulated gastrointestinal conditions.

·         The conclusion section looks like “another summary”, please modify it accordingly.

·         Moreover, in the conclusion section, you stated at the end that “…in line with the trend towards sustainable development and use of by-products, the development of a probiotic fermented milk with dehydrated cashew by-product has proven to be a promising alternative as a healthy food product”. Nevertheless, in one or two sentences, please include more accurate information about your future work related to this research?

Author Response

We thank the reviewers for their comments and suggestions. We are submitting a revised version of the manuscript with substantial modifications. In order to facilitate the visualization of changes and to ensure that each reviewer can easily identify modifications related to their suggestions, we have decided to mark the revisions suggested by Reviewer #1 in green and the ones suggested by Reviewer #2 in blue.

Reviewer 2

The abstract section needs to be improved, specially at the end, with more accurate information about the conclusion of your research. Moreover, the conclusion section of the manuscript is very similar that the abstract, therefore, please modify them.

  1. We have made significant modifications to the abstract, and we have rewritten the conclusion as per the request of both reviewers, and we have highlighted these modifications in yellow.

  • In line 29, in the keywords section. Since some words are repeated in the manuscript title and in the keyword section, I suggest to replace the words: “probiotics”, “by-product”, “cashew”, or “fermented milk”, by “antioxidant activity”, “fermentability assay”, “physicochemical characterization”, “fatty acid profile” or other keywords, to have more visibility of your manuscript paper.
  1. We found some of the suggestions, such as "antioxidant activity" and "fermentability assay," interesting. However, we believe that "physicochemical characterization" and "fatty acid profile" are not as suitable or appealing. We decided not to include the latter ones, and on the other hand, to include the term "in vitro digestion."
  • In the Introduction section, please, improve it including more information related to: i) the diverse chemical composition of fruit by-products; ii) fermentability assay, and iii) cultures employed, and the fermented milk production.
  1. The introduction has been enhanced with the suggested information.
  • In the Statistical Analysis section, did you the Levene's tests were performed for equality of variances and Shapiro-Wilk's tests to evaluate normal distribution. Nevertheless, what about the analysis of “autocorrelation in residuals”? Did you apply it? Why not?
  1. Among parametric, time serial data, in which regressions in the form of ANOVA were applied we evaluated residual plot as displayed by the Minitabâ‚¢ statistical suite. Notably, residuals were observed inside the fitted model inference, and as the fitted model was not used to forecast time-dependent trends, rather, its application was directed towards the comparative observation of testing versus control. Furthermore, the temporal series under consideration is constrained by a modest number of measurements conducted within the same monthly interval. In light of this, we thought it prudent to abstain from pursuing autocorrelation testing. Should we venture into predicting and understanding long-term colonization by the tested microorganisms due to fermented beverage consumption then it becomes a relevant test whether we can infer colonization by appropriate consumption.
  • Please, improve the discussion section of all your results including more information related to: i) Physical-chemical and functional characteristics of the cashew by-product, ii) Probiotic selection, iii) Viability of microorganisms used in fermented milk during storage, pH and titratable acidity, iv) Physical-chemical composition of fermented milk, and v) Survival of microorganisms in fermented milk under in vitro-simulated gastrointestinal conditions.
  1. The discussion has been improved in various aspects and in the topics indicated by the reviewer.
  • The conclusion section looks like “another summary”, please modify it accordingly.
  1. We have completely rewritten the conclusion section, as also suggested by the other reviewer, which is why it is highlighted in yellow.

  • Moreover, in the conclusion section, you stated at the end that “…in line with the trend towards sustainable development and use of by-products, the development of a probiotic fermented milk with dehydrated cashew by-product has proven to be a promising alternative as a healthy food product”. Nevertheless, in one or two sentences, please include more accurate information about your future work related to this research?
  1. Indeed, the investigations into the functional properties of the formulations developed here do not conclude with this manuscript. Future analyses will be undertaken, as indicated in yellow at the end of the conclusion.

Round 2

Reviewer 1 Report

The authors have revised accordingly

Reviewer 2 Report

Manuscript accepted in present form.